# The Role of Anthropogenic Elements in the Environment for Affective States and Cortisol Concentration in Mountain Hiking—A Crossover Trial

**DOI:** 10.3390/ijerph16020290

**Published:** 2019-01-21

**Authors:** Martin Niedermeier, Carina Grafetstätter, Martin Kopp, Daniela Huber, Michaela Mayr, Christina Pichler, Arnulf Hartl

**Affiliations:** 1Department of Sport Science, University of Innsbruck, Fürstenweg 185, 6020 Innsbruck, Austria; martin.kopp@uibk.ac.at; 2Institute of Ecomedicine, Paracelsus Medical University, Strubergasse 21, 5020 Salzburg, Austria; c.grafetstaetter@pmu.ac.at (C.G.); daniela.huber@stud.pmu.ac.at (D.H.); michaela.mayr@pmu.ac.at (M.M.); christina.pichler@pmu.ac.at (C.P.); Arnulf.Hartl@pmu.ac.at (A.H.)

**Keywords:** green exercise, cortisol, allostatic load, nature relatedness, anthropogenic elements, stress

## Abstract

Green exercise might have positive effects on health and affective states. Little is known about the ideal characteristics of the natural environment, where exercise is conducted in. Thus, the primary aim of the present study was to investigate the effects of anthropogenic elements on acute stress-related physiological responses and affective states in green exercise. Using a crossover field study design, 52 healthy participants were exposed to two different mountain hiking conditions: An environment with less anthropogenic elements and an environment with more anthropogenic elements. Pre and post conditions, affective states and salivary cortisol concentration were measured. Repeated measures ANOVAs were used to analyze if pre-post changes differed between the conditions. Pre-post changes in affective states and salivary cortisol concentration did not significantly differ, partial η² < 0.06. Positive affective states showed significantly higher values post compared to pre-condition, partial η² > 0.13. The present results indicate that anthropogenic elements have a minor role in the influence on affective states and salivary cortisol concentration during mountain hiking. It is concluded that a single bout of mountain hiking independent of anthropogenic elements in the environment is effective in influencing affective states positively.

## 1. Introduction

Physical activity is considered to be an important contributing factor to human health. It is recommended not only as an intervention when a chronic disease is present [1], but also—and maybe more importantly—as a prevention against diseases and for maintaining a healthy lifestyle [2]. At the same time, 27.5 % of the worldwide population showed insufficient physical activity in 2016 [3]. Therefore, a deeper understanding of the mechanisms to influence physical activity behavior is highly needed.

There is evidence that positively influenced affective state during and after physical activity may increase future physical activity behavior [4,5]. Consequently, scientific interest in factors influencing affective state, understood as an emotional state cognitively accessed [6], related to physical activity has increased in the past. Green exercise, defined as physical activity in a natural environment [7], was reported to be more effective in improving affective states compared to indoor exercise in a review [8]. The authors of the review summarized nine studies using various affective states measures (dimensional and categorical states) and concluded that green exercise enhanced positive affective states (e.g., positive engagement, pleasure) and decreased negative affective states (e.g., anger, depression) more effectively compared to indoor exercise [8]. When physical activity in urban versus natural environments is compared, there seems to be a more beneficial effect on affective states of physical activity in natural environments [9]. Repeatedly, these findings have been explained by a *positive* influence of specific stimuli in the natural environment. Two important theories for explaining possible benefits related to exposure to natural environments are the stress reduction theory (SRT) and the attention restoration theory (ART). In ART, benefits are related to the natural stimuli (so-called “soft fascinations”, which capture attention without effort, e.g., clouds, or moving trees), that create the feeling of “being away” leading to attention restoration accompanied with improved affective state [10]. In SRT, natural stimuli create a reduced complexity of the natural environment compared to an urban environment resulting in psychophysiological stress reduction including improved affective state [9]. It should be noted that these stimuli can be of various sensory impressions, e.g., visual, auditory, tactile, or olfactory perception [11]. Since the natural environment must be considered as diverse in itself, it remains unclear which elements of the natural environment are responsible for natural stimuli. Relatively few field studies addressed this question in green exercise research. Gidlow, et al. [12] compared exercise in green and blue (water present) to in green only environments, and hypothesized that the presence of water might be an important element to influence several outcomes including affective states. The authors did not report affective differences when exercising in green and blue and green only environments. Similarly, Rogerson, et al. [13] reported comparable improvements in affective states in different natural environments, i.e., beach, riverside, heritage, and grasslands. According to Tyrväinen, et al. [14] the effects of exercise in a park and a forest area on affective states were comparable, although both park and forest area produced superior results compared to an urban area.

However, both an indoor and an urban environment contain a high proportion of anthropogenic elements. By the term anthropogenic elements, we refer to elements constructed by humans (e.g., buildings, walls, or cars). Therefore, the findings on green exercise might be also explained by a *negative* influence of anthropogenic elements on affective states during exercise (opposed to a positive influence of specific stimuli of the natural environment). Mountain hiking is walking in mountainous areas with altitude differences [15]. Consistent with the green exercise literature, mountain hiking in a green outdoor environment produced larger improvements in affective states compared to treadmill walking indoor [15]. Specifically, affective valence, activation, and fatigue were more positively influenced in the green outdoor environment. Mountain hiking can be conducted both in environments with more anthropogenic elements (e.g., ski lifts, highway, or buildings) and in environments with less anthropogenic elements. The natural environment in mountain hiking plays an important role, since being close to nature is a main motive in engaging in mountain hiking [16]. Walking in mountainous areas was used earlier to study restorative effects of the environment on attention, emotion, and blood pressure [17]. Thereby, mountain hiking can be considered an appropriate form of physical activity to study possible influences of anthropogenic elements on affective states.

Although external influences (e.g., due to the environment) show an important impact on affective states, there are inconsistent indications that the effect of green exercise shows inter-individual differences related to personal traits. Specifically, the effect of green exercise might be influenced by nature relatedness, a trait considered as the individual level “of connectedness with the natural world” [18]. Rogerson, Brown, Sandercock, Wooller and Barton [13] reported a larger improvement in affective states in individuals with higher nature relatedness after a 5 km run. However, Gidlow, Jones, Hurst, Masterson, Clark-Carter, Tarvainen, Smith and Nieuwenhuijsen [12] could not reproduce these results for a 30 minute walking bout. Given the observation that the motive of being close to nature is important in mountain hiking, nature relatedness might play a role in environmental influences on affective states during mountain hiking [16].

In addition to affective states, acute physiological responses related to stress are an outcome of green exercise studies. Stress markers (e.g., cortisol concentration) are used to quantify the physiological stress status and possible stress relief responses [19]. Insufficient recovery from stress is considered a risk factor for cardiovascular diseases, diabetes and mental disorders [20]. Therefore, interventions to reduce stress and measures to support recovery from high levels of stress are of utmost concern. The evidence of additional effects, due to the environment in green exercise on acute stress-related physiological responses, is weak. Bowler, et al. [21] reported a non-significant effect size of Hedges *g* = 0.03 for cortisol concentration. However, knowledge about possible influences of the environment, where exercise is conducted, on cortisol concentration would help to optimize exercise recommendations.

Following these considerations, the primary aim of the present study was to analyze effects of anthropogenic elements in the natural environment during a single bout of green exercise (mountain hiking) on affective states and salivary cortisol concentration. In the context of the literature, we hypothesized that green exercise in an environment with less anthropogenic elements might show more favorable effects on affective states compared to green exercise in an environment with more anthropogenic elements.

The secondary aim was to identify the role of nature relatedness in the possible effects of anthropogenic elements. We hypothesized that persons with higher nature relatedness might benefit to a greater extent from green exercise in an environment with less anthropogenic elements compared to persons with lower nature relatedness.

## 2. Materials and Methods 

### 2.1. Participants

Participants were recruited by public announcements on the web page of the Austrian Alpine Association. On a first-come, first-serve basis, the first 52 participants without exclusion criteria were selected to participate. Exclusion criteria assessed by self-report were: (a) Pregnancy, (b) breast-feeding, (c) chronic or acute (in particular cardio-respiratory) diseases (already existing or diagnosed during the study), (d) age below 18 and above 65 years, (e) unable to be physically active assessed by the Physical Activity Readiness Questionnaire [22], (f) inability to speak and understand German. Incentives (e.g., backpacks, books, topographic maps) were provided for the participation in form of a raffle at the end of the study.

An a priori power analysis was performed to estimate the appropriate sample size for the study using G*Power version 3.1 (University of Düsseldorf, Germany) [23]. We hypothesized that the effects of anthropogenic elements in a natural environment might be comparable with the effects studied by Tyrväinen, Ojala, Korpela, Lanki, Tsunetsugu and Kagawa [14]. Using a similar study design, an effect size of *r* = 0.43 for the dimension of positive affect in the time by condition interaction was reported when a 30 minute bout of walking was conducted in a city environment compared to a park environment in the city. The effect size was converted to *f* using the conventions of [24,25] resulting in a *f* of approximately 0.19. Furthermore, the following assumptions and settings were used: α = 0.025, power = 0.80, dropout rate: 10 %, options: as in SPSS. Since two dimensional affective states were assessed, alpha was adapted using Bonferroni correction. Thereby, a sample size of *n* = 51 participants was calculated for detecting a significant condition by time interaction using a two × two fully repeated measures ANOVA.

### 2.2. Design 

The present field study consisted of an enrollment phase and an experimental phase in a crossover design (Figure 1). Initial screening was done prior to the experimental phase via a web-based questionnaire, where sociodemographic data, nature relatedness, perceived stress, and physical activity readiness was collected. The experimental phase was conducted on three consecutive days in October 2017 (mid-autumn) and was located in a renowned summer and winter sports area in Austria. All participants arrived on the first day at the hotel, were exposed to two experimental conditions on the second and the third day, and departed on the third day. On the first day, the total sample was divided into two groups. One group started with mountain hiking in an environment with less anthropogenic elements (N) on the second day and continued with mountain hiking in an environment with more anthropogenic elements (H) on the third day. The other group started with H the second day and continued with N on the third day. The allocation process to the groups was not randomized in order to avoid splitting related participants (e.g., families, spouses). The participants were informed that two mountain hiking tours will be conducted. The details of the two conditions and the study question were not communicated to the participants to avoid cognitive bias (e.g., positive responses to N). Each group was accompanied by three researchers throughout the mountain hiking tours. The team of researchers per group did not change between the two days to minimize social influences by the researchers. The study was approved by the Board for Ethical Questions in Science of the University of Innsbruck (number: 28_2017) and all participants signed a consent form after obtaining written and spoken information about the study procedures. Pre-registration of the study was done at ClinicalTrials.gov (identification number: NCT03229434, July 2017).

### 2.3. Procedure 

The procedure of the conditions on the second and third day was identical with the exception of the location where the mountain hiking tours were conducted (Figure 2). Data were collected on four different time points: In the morning after wake-up (morning), immediately prior to the mountain hiking tours (pre), at the turning point (mid), and immediately after the mountain hiking tours (post). During the mountain hiking tours, all assessments, brakes, the start, and the end were synchronized between the two research teams using mobile phones to guarantee the identical timeline in both groups. Affective states were measured two times and salivary cortisol concentration was assessed three times. Food intake was not allowed 60 minutes before saliva collection. All participants rinsed out their mouth with clean water 10 minutes prior to saliva collection. Heart rate was measured continuously during the mountain hiking tours. Activities of the participants after returning to the hotel and debriefing were not standardized.

### 2.4. Conditions

The two mountain hiking tours were carefully selected prior to the study start by the research team with the following demands: As similar as possible in terms of physical demands (distance, altitude meters, and altitude profile) and distinctive in terms of the environment (more and less anthropogenic elements). The total distance was approximately 7 km and contained 750 altitude meters for both mountain hiking tours each. Total time for each hiking tour was approximately three hours, where the uphill hiking phase lasted approximately 100 minutes and downhill hiking phase around 70 minutes. The same track was used for uphill (average speed: 2.1 km/h) and downhill hiking (average speed: 3 km/h).

The mountain hiking tour in the environment with more anthropogenic elements (H) started at 1050 m and ended at 1800 m above sea level. Anthropogenic elements were present all along the tour (e.g., highway, ski lift, snow cannons, construction sites, cars at the starting point, Figure 3). The path quality mainly consisted of forest roads and gravel roads. The turning point was at the top station of a ski lift.

The mountain hiking tour in the environment with less anthropogenic elements (N) started at 1600 m and ended at 2350 m above sea level. All mentioned anthropogenic elements for H were not existent in N (Figure 3). The path quality was predominantly small paths and pathless terrain. The turning point, as well as the main part of the tour was without any buildings in sight.

### 2.5. Measurements

#### 2.5.1. Initial Screening

Information on the ability to be physically active, sociodemographic data, physical activity, nature relatedness and perceived stress was collected in a web-based questionnaire with 74 questions (total time to complete: Approximately 30 minutes). The six-item Physical Activity Readiness Questionnaire [22] was used for assessing the ability to be physically active. The short form of the International Physical Activity Questionnaire consisting of 11 items was used for assessing self-reported physical activity (http://www.ipaq.ki.se/). The questionnaire investigates the frequency and duration of vigorous, moderate, and walking activity during the last seven days, as well as the sitting time per day in 11 items. Physical activity is expressed by energy expenditure in metabolic equivalent minutes (MET min). Nature relatedness was assessed using the Nature Relatedness Scale (NRS) [18]. The scale consists of 21 items to be answered on five-point Likert scale ranging from “disagree strongly” (1) to “agree strongly” (5) and allows calculating a total mean score for nature relatedness ranging from low nature relatedness (1) to high nature relatedness (5). Perceived stress was assessed using the Perceived Stress Scale [26], where 10 items are rated on a five-point Likert scale ranging from “never” (0) to “very often” (4). The total sum score allows a quantification of the appraisal of feelings of stress in the last month and ranges from 0 (low perceived stress) to 40 (high perceived stress).

#### 2.5.2. Affective States

Four different sets of self-report questionnaires based on the Circumplex model [6] were used to assess affective states. Two single-item scales were used to cover the two dimensions affective valence (pleasure) and arousal: The Feeling Scale by Hardy and Rejeski [27] assesses affective valence and consists of 11 answer possibilities ranging from “very good” (+5) to “very bad” (−5) with a neutral answer possibility. Values for convergent validity are reported to range from r = 0.41 to 0.88 [28]. The dimension of arousal was assessed using the Felt Arousal Scale [29]. The scale provides six response options ranging from “low arousal” (1) to “high arousal” (6) without a neutral answer possibility. Van Landuyt, Ekkekakis, Hall and Petruzzello [28] reported convergent validity with the arousal dimension of the Self Assessment Manikin [30] or the Arousal Scale of the Affect Grid [31] (r = 0.45 to 0.70). The combination of Feeling Scale and Felt Arousal Scale is frequently used in exercise psychology (e.g., Ekkekakis, et al. [32]).

Two multi-item scales were used to cover nine selected distinctive categorical affective states: The Mood Survey Scale is an adjective list of 40 items (“At this moment, I feel…”) with a five-point Likert response mode (“not at all” to “very”) [33]. The sum of five items is used to calculate eight subscales (activation, elation, contemplation, calmness, fatigue, depression, anger, excitement) ranging from five (lowest value) to 25 (highest value). The subscales can be located in the Circumplex model: Activation and excitement mark affective states with high arousal. Activation is characterized by more positive valence and excitement with more negative valence. The subscales fatigue (negative valence) and calmness (positive valence) mark low arousal. Anger, depression (both negative valence), contemplation (neutral valence) and elation (positive valence) all mark medium aroused states [33]. Internal consistency for the subscales ranges between Cronbach’s α = 0.70 and 0.88 [33]. Psychometric properties to convergent and divergent validity are provided by Abele-Brehm and Brehm [33]. The German version of the State Trait Anxiety Inventory was used to assess the dimension of state anxiety [34,35]. The 20 items of the subscale state anxiety are summed up to calculate a score ranging from 20 (low anxiety) to 80 (high anxiety). In the Circumplex model, anxiety is described with higher arousal and more negative valence [6]. The German state subscale shows Cronbach’s α between 0.90 and 0.94 indicating high internal consistency [35].

After the tours, the participants were asked on the belief about the influence of the environment on well-being using a single item question (“How did the environment of the mountain hiking tour influence your well-being?”) with a response mode ranging from 1 (“100% negatively”) over 5 (“50% negatively, 50% positively”) to 10 (“100% positively”).

#### 2.5.3. Salivary Cortisol Concentration

Salivary cortisol concentration was used to quantify the physiological stress status of the participants. Passive drooling method was used to collect salivary samples (2.5–3.5 mL) in polypropylene vials, which were stored separately at −20°. Cortisol concentrations were determined collectively at the laboratory of Paracelsus Medical University Salzburg, Austria, where samples were thawed up. After centrifugation at 1500 × g for 10 min to separate mucins, samples were analyzed using a Cortisol Saliva ELISA Free (SA E-6000) assay from LDN^®^ (Labor Diagnostika Nord GmbH and Co. KG, Nordhorn, Germany) according to manufacturer’s instructions. The absorbance of each sample got determined at 450 ± 10 nm with a calibrated microtiter plate reader (Anthos Zenyth 3100, model DTX 880, Friesoythe, Germany). Concentrations were calculated using a four parameter logistics curve fit and were expressed in nmol/L.

#### 2.5.4. Walking Intensity

Due to a limited amount of available heart rate monitors, walking intensity was assessed by heart rate in a randomly selected subsample (*n* = 10) of all participants. Heart rate was continuously measured throughout the mountain hiking phases between pre and post using a heart rate monitor with an optical wrist sensor (M430, Polar Electro Oy, Finland). The identical participants wore the heart rate monitors in N (environment with less anthropogenic elements) and H (environment with more anthropogenic elements). Accuracy of the device is described with ± 1 beats per minute under stable conditions by the manufacturer. Heart rate values were averaged for N and H and were expressed as an estimated percentage of maximal heart rate using the formula of Tanaka, et al. [36].

### 2.6. Statistical Analyses

All statistical analyses were performed using SPSS version 24 (IBM, New York, United States). Affective states and salivary cortisol concentration were considered the main outcomes of the study. Possible differences prior to the conditions were tested by separate one-factorial analyses of variances with repeated measurements (ANOVAs) with condition as within-factor (N, H) on each outcome. For the primary analysis of affective states, a series of 2 × 2 fully repeated measures ANOVAs was used to analyze the effect of condition (N, H), time (pre, post) and condition by time interactions. Significant interactions between condition and time were considered as different changes in the parameters. In the secondary analysis, NRS group (low nature relatedness, high nature relatedness determined by median split), was added as a between-group factor to the ANOVAs resulting in 2 × 2 × 2 mixed-model ANOVAs. Significant 3-way interactions between condition, time, and group were considered as different influences of the environment in persons with low compared to persons with high NRS. It should be noted that median nature relatedness was high in the present sample (4.1) compared to previous studies on nature relatedness (mean value of environmentalists 3.7 [18]). Since our statistical analysis approach was determined a priori we continued with the analysis strategy. A similar approach was conducted for salivary cortisol concentration, which was assessed three times a day (opposed to two stages for affective states). Therefore, the factor time consisted of three stages (morning, mid, post) and the primary analysis was a 2 × 3 fully repeated measures ANOVA. The secondary analysis was a 2 × 2 × 3 mixed-model ANOVA. Square-root transformation of the salivary cortisol concentration values was conducted prior to analysis. Whenever the assumption of sphericity was not met in the ANOVA, Greenhouse-Geisser correction was applied. Partial η² (η²*p*) was used as an effect size.

The global significance level was set at α = 0.05 (two-tailed). Since multiple outcome measures for affective states were analyzed, Bonferroni correction was applied and resulted in *p*-values of 0.025 (dimensional affective responses) and 0.006 (categorical affective responses), respectively. Unless otherwise stated, data are presented as mean (standard deviation).

## 3. Results

### 3.1. Preliminary Analyses

Demographic data of the 52 participants are shown in Table 1 for the total sample and for the subgroups with low and high nature relatedness determined by a median-split of the sample. Participants with high nature relatedness reported higher physical activity, a higher prevalence of being single and being outdoors for more than three times per week. Significant differences prior to the mountain tours were found for all affective states, *p* < 0.040, η²*p* > 0.08, with the exception of fatigue and excitement, where no significant differences were found, *p* > 0.254. All affective states were rated more positively on the first day compared to the second day. No significant carryover effects were observed, *p* > 0.160. Walking intensity according to the average heart rate (*n* = 10) was comparable in N (63 (10) % of age-predicted maximal heart rate) and H (63 (7) % of age-predicted maximal heart rate), respectively. No harmful event and no dropout were observed over the experimental phase.

### 3.2. Affective States and Salivary Cortisol Concentration

For none of the affective states, a significant condition by time interaction was found, *p* > 0.083, η²*p* < 0.06, indicating similar pre-post changes in N and H for affective states (Table 2). Significant time effects were found for affective valence (according to the Feeling Scale), elation, excitement, and anxiety. Positive affective states (elation and affective valence) showed higher values post-condition compared to pre-condition, negative affective states (anxiety, excitement) showed lower values. A significant condition effect was found for arousal according to the Felt Arousal Scale, indicating a higher arousal in N compared to H.

The belief on how the environment influenced well-being showed a significant difference between N: 8.5 (1.7) and H: 6.3 (2.4), *p* < 0.001, η²*p* = 0.41, indicating a more positive influence of the environment during mountain hiking in N compared to H.

Salivary samples of eight participants could not be analyzed, resulting in *n* = 44 participants for the analysis of salivary cortisol concentration. There was a significant time effect indicating a decrease over time, *p* < 0.001, η²*p* = 0.73 (Figure 4). Both the factor condition, *p* = 0.937, η²*p* < 0.01, and the condition by time interaction, *p* = 0.683, η²*p* = 0.01, were not significant indicating comparable values and decreases over time in cortisol concentration in N and H.

### 3.3. Subgroup Analysis

None of the affective states showed a significant 3-way interaction (time by condition by group) according to the Bonferroni-corrected *p*-value of 0.006. In the affective state calmness, values of *p* = 0.007, η²*p* = 0.13, were evident. Participants with low nature relatedness reported an increase of calmness in H and a decrease of calmness in N (H pre: 18.0 (3.6), H post: 18.5 (3.7), N pre: 18.6 (2.3), N post 18.0 (2.7)). Participants with high nature relatedness reported an increase of calmness in N and a decrease of calmness in H (H pre: 20.2 (3.7), H post: 18.7 (3.0), N pre: 19.5 (3.2), N post 20.4 (2.7)). All other affective states, *p* > 0.071, η²*p* < 0.06, and salivary cortisol concentration did not show significant 3-way interactions, *p* = 0.081, η²*p* = 0.06.

## 4. Discussion

The main objective of the present study was to identify the influence of anthropogenic elements on affective states and salivary cortisol level during mountain hiking. The results suggest comparable changes of both affective states and salivary cortisol level when hiking in environments with less and more anthropogenic elements. Based on the results observed, we conclude that anthropogenic elements in the natural environment may play a minor role in the outcomes studied during a three-hour mountain hike.

### 4.1. Influence of Anthropogenic Elements on Affective States and Salivary Cortisol Concentration during Mountain Hiking

Exposure to the natural environment without physical activity was shown to have a positive influence on several psychological parameters including affective states [37]. It is believed that green exercise may have a “synergistic benefit” [38]. These additional benefits on affective states are important in relation to well-being and psychological health [21], but might also positively influence future physical activity [4]. Based on previously reported findings of indoor (as a proxy for a high proportion of anthropogenic elements) and urban exercise [8,9], we hypothesized a negative influence of anthropogenic elements on affective states, when exercising in a natural environment.

However, a negative influence of anthropogenic elements on affective states was not confirmed in the present study. Some affective states showed non-significant changes in the expected direction (e.g., elation showed a larger increase after hiking in an environment with less anthropogenic elements compared to more anthropogenic elements), but the effects of anthropogenic elements during physical activity must be considered as small when compared to the contrast between indoor and outdoor activity [8] and as even smaller when compared to the effect of the physical activity during mountain hiking [15]. Put in the wider context of ART and SRT, it seems plausible that (still undetected) specific stimuli of the natural environment (opposed to a negative influence of anthropogenic elements) might explain green exercise benefits, since in the present study, the condition with more anthropogenic elements also contained natural stimuli. Tyrväinen, Ojala, Korpela, Lanki, Tsunetsugu and Kagawa [14] compared walking in an urban park area and walking in a city area and reported a significantly larger pre-post change of positive dimensional affective state in the urban park area. However, the authors also compared walking in an urban park area to walking in a forest area and reported a non-significant enhancing effect on positive affective state. We cautiously conclude that the contrasts in the environmental characteristics (between the environment containing more and less anthropogenic elements) in the present study are more comparable with differences between an urban park area and forest area (and less with differences between urban park area and city area) studied in Tyrväinen, Ojala, Korpela, Lanki, Tsunetsugu and Kagawa [14]. Another possible explanation for the lack of favorable effects in the environment with less anthropogenic elements might refer to the anthropogenic elements ski lifts and snow cannons. In winter time, potentially joyful activities like skiing and snowboarding are realized by ski lifts and snow cannons. Consistent with mood congruency [39], recall on previous joyful activities might have influenced affective responses in the present study.

Interestingly, this lack of influence of the anthropogenic elements seems to hold true for the acute assessment of the affective states only. When asked for the belief about the influence of the environment, the environment with less anthropogenic elements was rated significantly more positively compared to the environment with more anthropogenic elements. This indicates a discrepancy between the acute assessment of affective states and the belief about the influence of the environment. The psychological outcomes used in the present study (affective states) were selected because of the recommendations of Ekkekakis and Petruzzello [40], as well as their relevance to future physical activity [4]. However, according to the Reasoned Action Approach, also the beliefs (influencing the attitudes towards the behavior) may change physical activity behavior [41]. Therefore, it seems to be important to include validated belief- and attitude-based assessment instruments in future research of possible environment-based add-on-effects of exercise. This is also suggested based on the finding that being close to nature is an important motive in engaging in mountain activities [16].

The only significant condition effect was found for arousal, indicating that independent of the time, arousal was rated higher in the condition with less anthropogenic elements. This was surprising on the first sight, since according to previous findings [9], it was expected that the characteristics of the green environment will result in lower arousal. However, an important characteristic of mountain hiking must be taken into account to correctly interpret the higher arousal. Mountain hiking is associated with a certain risk of accidents where the majority of the accidents (> 90%) occur on small paths and pathless terrain [42]. Consequently, the higher arousal in the condition with less anthropogenic elements might reflect increased attention, triggered by the poorer path quality.

Salivary cortisol concentration showed a comparable decrease during mountain hiking in environments with less and with more anthropogenic elements. This result is congruent with the findings of walking/running in synthetic and outdoor environments summarized in Bowler, Buyung-Ali, Knight and Pullin [21]. The authors reported that there is “less evidence of a consistent difference between environments” in various physiological variables including cortisol concentrations. Similarly, we did not detect different changes in salivary cortisol concentration between mountain hiking indoor and outdoor in one of our previous studies [43]. It has to be noted that environmental differences (due to pleasant and unpleasant scenes) were shown for other stress-related physiological parameters (blood pressure) previously in a laboratory study, when exercising in front of pictures on a treadmill [7]. Thereby, it might be concluded that such effects might not be detected due to multiple influencing factors in a field study.

### 4.2. Positive Time Effects on Affective States and Salivary Cortisol Concentration

Regardless of the environment, affective states were positively influenced by mountain hiking. Specifically, both dimensional affective valence (pleasure) and categorical state elation increased from pre to post mountain hiking. Negative affective states (excitement and anxiety) were rated lower after the mountain tour compared to before hiking. These results are in line with the literature on mountain hiking. In our previous studies, we also found significant increases of affective valence and elation and a decrease in anxiety [15]. In this previous study, the changes in mountain hiking were compared to a sedentary control condition, what might be considered as well-controlled. The sedentary control condition might also explain some discrepancies to the present study, where no significant change in activation was found. In Niedermeier, Einwanger, Hartl and Kopp [15], the significant interaction in activation was mainly driven by a decrease of activation in the control condition (mean change: −2.3), while activation in mountain hiking stayed relatively stable (mean change: 0.6).

The significant decrease in salivary cortisol concentration over time may be explained to a large amount by the circadian rhythm [44]. However, taking our previous findings of mountain hiking into account, the circadian rhythm can be additionally influenced by the physical activity during mountain hiking leading to a larger salivary cortisol concentration [43]. Thereby, a single bout of mountain hiking (independent of environmental characteristics) might be recommended as an intervention to reduce salivary cortisol concentration and in the longer run as prevention against cardiovascular diseases, diabetes and mental disorders [20].

### 4.3. Nature Relatedness

As a secondary aim of the study, a possible influence of nature relatedness on the environment-green exercise relation was tested. Based on the non-significant findings in the subgroup analysis, the main conclusion has to be that nature relatedness is a less important variable to consider. This may support the results of Gidlow, Jones, Hurst, Masterson, Clark-Carter, Tarvainen, Smith and Nieuwenhuijsen [12], who reported “no potential relationship linking nature-relatedness with restorative experience“ for a 30 minute walking bout. However, this conclusion is challenged by at least two factors. Firstly, the mean values for nature relatedness of both subgroups (low nature relatedness: 3.8, high nature relatedness 4.4) are above the mean value of environmentalists of 3.7 reported previously [18]. Thereby, the present sample is considered a homogenous group with high nature relatedness, what might have concealed possible influences of nature relatedness. Secondly, the *p*-values of the analysis were Bonferroni-corrected, what increases the probability of a type-II error [45]. In the subscale calmness, the descriptive values showed a (non-significant) comprehensible development (in the environment with less anthropogenic elements: Increased calmness in participants with high nature relatedness opposed to decreased calmness in participants with low nature relatedness), but the critical *p*-value was missed by 0.001. Thereby, the possible influence of nature relatedness requires further research.

### 4.4. Limitations

The following limitations have to be considered when interpreting the findings: Firstly, a selection bias might be present given both the recruitment process (homepage of Austrian Alpine Association) and the high values in nature relatedness. Thereby, the conclusions drawn should not be generalized to other populations. Secondly, the reported discrepancies between the acute assessment of affective states and the belief about the influence of the environment are based on a non-validated single-item scale and have to be confirmed in future studies. Thirdly, the conceptualization as a field study decreases internal validity by introducing immutable factors (e.g., weather). In the present study, the weather conditions were different between the first hiking tour (mostly sunny) and the second hiking tour (mostly cloudy and rainy) what resulted in significant differences in most affective states prior to the tour. Our analyses of the pre-post change values (opposed to the post values) are believed to minimize the influence of the different weather conditions; however, we cannot exclude ceiling or floor effects. Fourthly, we did not control for social interaction between the participants what might have affected the results.

## 5. Conclusions

The present field study focused on a possible impact of anthropogenic elements in the environment on affective states and salivary cortisol level in a single three-hour bout of green exercise (mountain hiking). According to the present results, we conclude that anthropogenic elements in the natural environment seem to be a minor influencing factor on the change of the outcomes observed. The role of nature relatedness remains unclear and future research is warranted.

## Figures and Tables

**Figure 1 ijerph-16-00290-f001:**
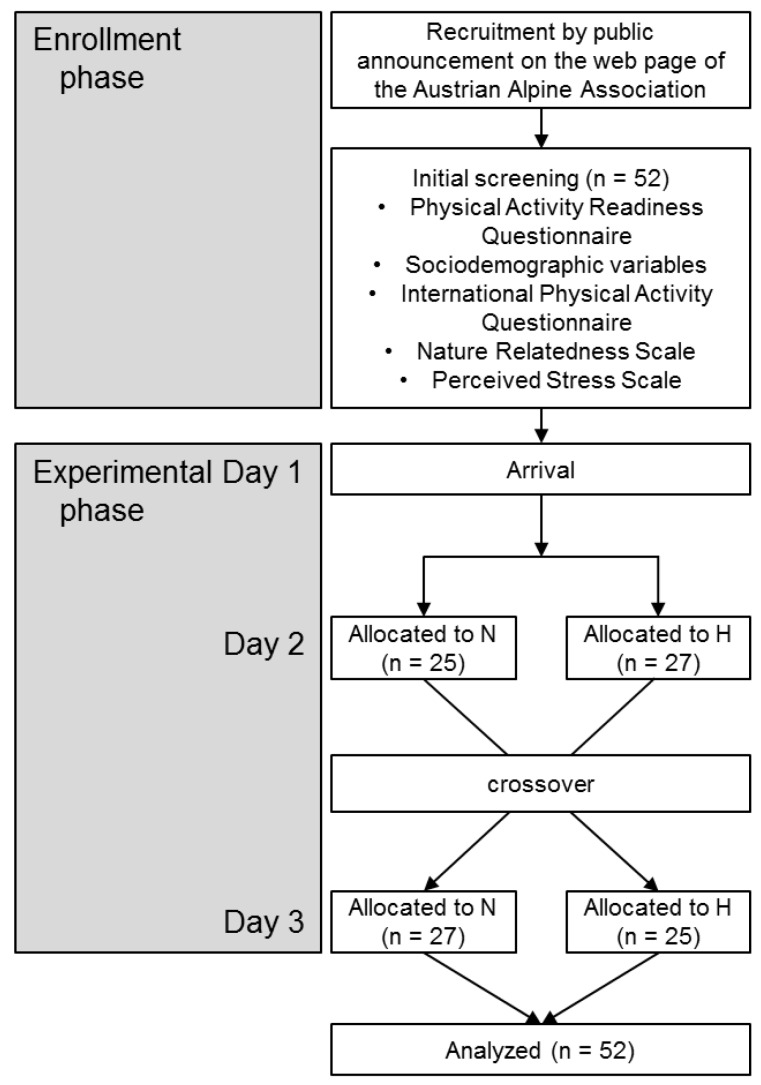
Flow diagram for data collection. N: Hiking in an environment with less anthropogenic elements, H: Hiking in an environment with more anthropogenic elements.

**Figure 2 ijerph-16-00290-f002:**
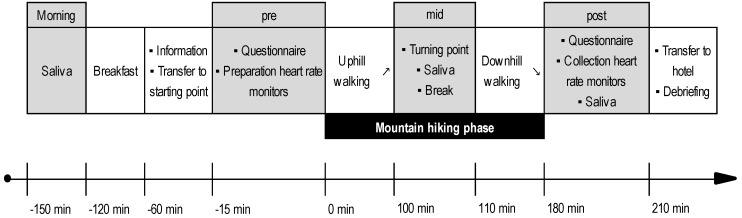
Experimental procedure for one condition. Grey fields mark data collection time points.

**Figure 3 ijerph-16-00290-f003:**
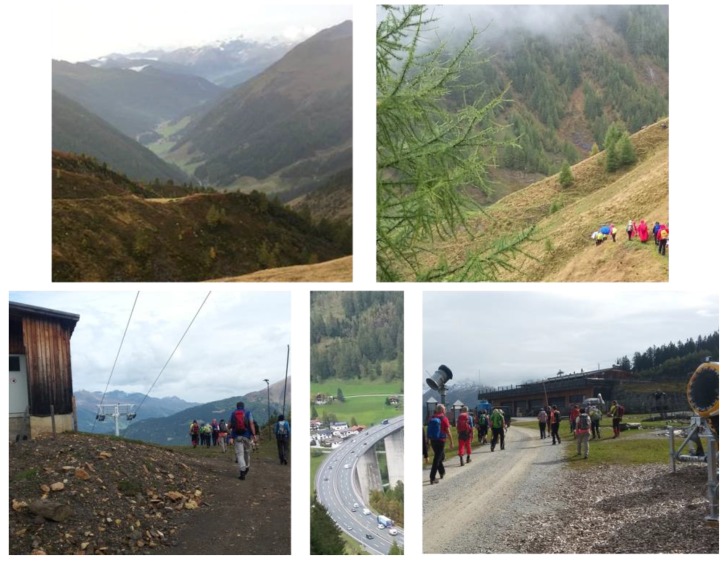
Typical view while hiking in the environment with less anthropogenic elements (N, two upper pictures) and the environment with more anthropogenic elements (H, three lower pictures).

**Figure 4 ijerph-16-00290-f004:**
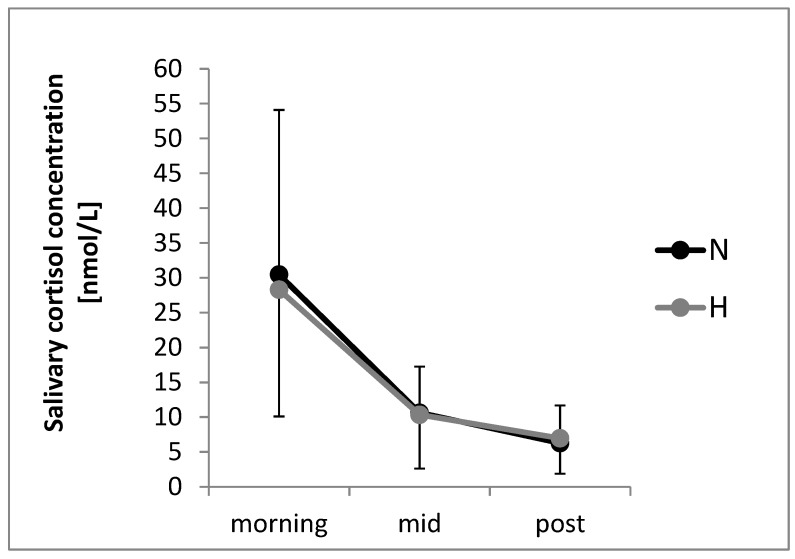
Mean salivary cortisol concentration for each time point by conditions. N: Environment with less anthropogenic elements, H: Environment with more anthropogenic elements. Error bars represent standard deviations. Missing data: *n* = 8.

**Table 1 ijerph-16-00290-t001:** Demographic data for the total sample and by subgroups with low and high nature relatedness.

Variable	Total Sample (*n* = 52)		Subgroups			
			Low NRS ^a^ (*n* = 27)		High NRS ^a^ (*n* = 25)	
	Mean	(SD)	Mean	(SD)	Mean	(SD)
Age (years)	47	(13)	44	(12)	49	(13)
Body mass index (kg/m2)	23	(4)	24	(5)	22	(3)
Mountain tours (n/year)	40	(47)	36	(40)	44	(55)
Physical activity (MET mins ^b^ per week) ^c^	6063	(4927)	4523	(2585)	7541	(6124)
Nature Relatedness (1: low, 5: high)	4.1	(0.4)	3.8	(0.3)	4.4	(0.2)
Perceived Stress (0: low, 40: high)	13	(5)	13	(6)	13	(5)
	%	(*n*)	%	(*n*)	%	(*n*)
Sex, female ^d^	57.7	(30)	55.6	(15)	60.0	(15)
Relationship status, single ^d^	27.5	(14)	15.4	(5)	40.0	(10)
Being outdoors 3 times and more per week	59.6	(31)	44.4	(12)	76.0	(19)
Being outdoors in nature, e.g. nature reserve, 3 times and more per week	13.5	(7)	11.1	(3)	16.0	(4)

^a^: NRS: nature relatedness, ^b^: MET mins: metabolic equivalent minutes, ^c^: missing data *n* = 3, ^d^: missing data *n* = 1 each.

**Table 2 ijerph-16-00290-t002:** Mean (SD) values for affective states by condition and time points.

	N ^a^		H ^b^		*p*-Value			η²*p* ^c^		
Affective States	pre	post	pre	post	Condition	Time	Interaction	Condition	Time	Interaction
Affective valence	2.7 (2.2)	3.6 (1.5)	2.7 (2.1)	3.3 (1.8)	0.497	**<0.001**	0.445	0.01	**0.26**	0.01
Arousal	3.0 (1.1)	2.9 (1.6)	2.7 (1.4)	2.5 (1.1)	**0.011**	0.311	0.946	**0.12**	0.02	0.00
Activation	17.5 (4.2)	17.1 (3.6)	17.9 (3.9)	16.7 (3.0)	0.940	0.071	0.245	0.00	0.06	0.03
Elation	18.1 (4.6)	19.9 (3.6)	18.1 (4.4)	18.7 (3.7)	0.320	**0.006**	0.124	0.02	**0.14**	0.05
Calmness	19.0 (3.5)	19.1 (2.9)	19.0 (3.8)	18.7 (2.7)	0.593	0.826	0.446	0.01	0.00	0.01
Fatigue	9.0 (4.2)	7.9 (3.3)	8.4 (3.8)	7.8 (3.1)	0.268	0.075	0.388	0.02	0.06	0.01
Depression	6.6 (2.4)	6.5 (2.7)	7.2 (3.5)	6.8 (3.0)	0.209	0.368	0.702	0.03	0.02	0.00
Contemplation	11.4 (3.3)	10.8 (3.3)	10.4 (2.9)	10.7 (3.6)	0.143	0.629	0.083	0.04	0.00	0.06
Anger	6.8 (2.9)	6.6 (2.8)	6.9 (3.8)	7.1 (3.6)	0.390	0.978	0.577	0.01	0.00	0.01
Excitement	7.8 (2.5)	6.9 (2.1)	8.1 (2.7)	7.2 (2.7)	0.119	**0.003**	0.951	0.05	**0.17**	0.00
Anxiety	35.6 (7.2)	32.9 (6.2)	35.8 (8.5)	34.2 (6.0)	0.321	**0.005**	0.406	0.02	**0.14**	0.01

^a^: N: Hiking in an environment with less anthropogenic elements, ^b^: H: Hiking in an environment with more anthropogenic elements, ^c^: η²*p*: Effect size partial η squared. Bold values indicate significant *p*-values/effect sizes.

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
