# Peer review of "The Role of Anthropogenic Elements in the Environment for Affective States and Cortisol Concentration in Mountain Hiking—A Crossover Trial"

_ijerph, 2019, doi:10.3390/ijerph16020290_

Round 1

Reviewer 1 Report

Theme, psysiological and psychological effects of mountain hiking is interesting.

But there are some fundamental deficits.

1,

I can’t understand the definition of more anthropogenic elements.

The physiological effects of mountain hiking are affected not only by landscape, visual stimulus, but also by olfactory stimulation, hearing, and tactile stimulation.

For example, see the paper Lanki, etc. Acute effects of visits to urban green environments on cardiovasvular physiology in women: A field experiment. Environ. Res., 2017, 176-185.

Definition of stimulation and physical measurement values are necessary.

2,

Cortisol affects diurnal variation and exercise effect.

Both of them do not separate in this research design, the effect of mountain hiking is not clear.

For example, it is conceivable that the diurnal variation or the exercise effect is greater than the influence of visual artifacts.

Author Response

At first, we want to thank the Reviewers for their time spent on the evaluation of the manuscript. We believe the comments greatly helped to improve the manuscript.

Please find the comments of the Reviewers in black and the author’s responses in blue.

Reviewer #1

English language and style: Moderate English changes required

We followed the suggestion of the Editor and the Reviewer and critically re-checked the manuscript and adapted, where necessary.  

Theme, psysiological and psychological effects of mountain hiking is interesting.

But there are some fundamental deficits.

1,

I can’t understand the definition of more anthropogenic elements.

Thank you for that comment. Our intention was using a term that summarizes characteristics in the environment that are constructed by humans and that are opposed to elements considered as “green” or “natural” (a term already used by Ulrich et al., 1991). Previously used terms (e.g. urban environment (Ulrich et al., 1991), synthetic environment (Bowler et al., 2010), indoor environment (Thompson Coon et al., 2011)) did not seem appropriate for the present study since mountain hiking is usually conducted in a natural environment. However, the mountain hiking natural environment differs by the amount of anthropogenic elements. As we are not aware of an already stated definition of the term “anthropogenic elements”, we added a sentence on the first appearance of anthropogenic elements early in the manuscript.

“By the term anthropogenic elements, we refer to elements constructed by humans (e.g. buildings, walls, or cars).”

Furthermore, we included an additional figure to show the differences between the environment with less anthropogenic elements (N) and the environment with more anthropogenic elements (H) more clearly.

Figure 3. Typical view while hiking in the environment with less anthropogenic elements (N, two upper pictures) and the environment with more anthropogenic elements (H, three lower pictures).

The physiological effects of mountain hiking are affected not only by landscape, visual stimulus, but also by olfactory stimulation, hearing, and tactile stimulation.

For example, see the paper Lanki, etc.,. Acute effects of visits to urban green environments on cardiovasvular physiology in women: A field experiment. Environ. Res., 2017, 176-185.

We fully agree with the Reviewer on the fact that the visual inputs of the landscape are not the only source for possible effects on physiological (and also psychological) parameters and that various sensory inputs may be responsible for these effects. This was insufficiently reflected by our description in the methods section “Anthropogenic elements were visible (and audible)…”. The comment of the Reviewer was assumedly triggered by the following misleading sentence in the discussion section. “This was surprising on the first sight, since according to previous findings [9], it was expected that the visual stimuli of the green environment will result in a lower arousal.”

Accordingly, we adapted the sentence as follows:

“This was surprising on the first sight, since according to previous findings [9], it was expected that the characteristics of the green environment will result in a lower arousal.”

Furthermore, we adapted the sentence in the methods section to reflect not only visible and audible elements but all potential sources of change:

“Anthropogenic elements were present all along the tour…”

Additionally, we want to thank the Reviewer for the citation that we included in the present version of the manuscript to explicitly state that the complexity of inputs can be of importance (opposed to only visual stimuli):

“It should be noted that these stimuli can be of various sensory impressions, e.g. visual, auditory, tactile, or olfactory perception [11].”

Definition of stimulation and physical measurement values are necessary.

As we did not use the term “stimulation” in the previous version of the manuscript, the Reviewer may refer to the term “stimuli”, which is used by both Ulrich et al. and Kaplan et al. We referred to the term stimuli, when we put the results into the context of the two theories, stress reduction theory and the attention restoration theory. We tried to clarify the term early in the manuscript by including the following sentence:  

“In ART, benefits are related to the natural stimuli (so-called “soft fascinations”, which capture attention without effort, e.g. clouds, or moving trees), that create the feeling of “being away” leading to attention restoration accompanied with improved affective state [10].“

We are not sure, if we understand the Reviewer correctly with the term “physical measurement values”. We used two physiological measures in the present study: salivary cortisol concentration as a measure of physiological responses related to stress and heart rate as a measure of waling intensity

To avoid misunderstandings, we included the following sentence to the sub-headline of Salivary Cortisol Concentration in the methods section and adapted the sentence in the discussion section:

“Salivary cortisol concentration was used to quantify the physiological stress status of the participants.”

“Thereby, a single bout of mountain hiking (independent of environmental characteristics) might be recommended as an intervention to reduce salivary cortisol concentration and in the longer run as prevention against cardiovascular diseases, diabetes and mental disorders”

2,

Cortisol affects diurnal variation and exercise effect.

Both of them do not separate in this research design, the effect of mountain hiking is not clear.

For example, it is conceivable that the diurnal variation or the exercise effect is greater than the influence of visual artifacts.

We agree that cortisol concentration is affected by both the diurnal variation and the exercise effect, which we discussed in the manuscript with the sentences of circadian rhythm and physical activity:  

“The significant decrease in salivary cortisol concentration over time may be explained to a large amount by the circadian rhythm [43]. However, taking our previous findings of mountain hiking into account, the circadian rhythm can be additionally influenced by the physical activity during mountain hiking leading to a larger salivary cortisol concentration [42]. Thereby, a single bout of mountain hiking (independent of environmental characteristics) might be recommended as an intervention to reduce salivary cortisol concentration and in the longer run as prevention against cardiovascular diseases, diabetes and mental disorders [19].”

Our previous study can be considered as a confirmation of the Reviewer’s assumption “the exercise effect is greater than the influence of visual artifacts”. Nevertheless, our intention was to study the effects of environmental influences in the (green) exercise setting based on previous findings (Pretty et al. 2005). If such effects were to be found, they might be considered of high practical importance since an add-on effect on diurnal variation and exercise would be existent.

However, as pointed out in the introduction section, it was less expected that we will observe changes in physiological parameters:

“The evidence of additional effects due to green exercise on acute stress-related physiological responses is weak. Bowler, et al. [20] reported a non-significant effect size of Hedges g = 0.03 for cortisol concentration.”

Reviewer 2 Report

Review of the manuscript «Is there an environment-based add-on-effect of mountain hiking on affective states and cortisol concentration? A cross over trail.

Thank you for the opportunity to read the manuscript. The manuscript has a good structure and provides a lot of useful information about the field study. I have a few comments that may help discussing their approach and improve the text.

The manuscript contributes to the discussion on how environmental qualities may promote or enhance psychological benefits of mountain hiking as a form of physical activity. The design and exposure conditions selected may also be viewed as a possibility to uncover to what extent anthropogenic components in the environment play a role for the psychological benefits of being active in nature. It is worth mentioning that mountain hiking was an exposure of interest in the early classic studies on nature exposure in environmental psychology (see Hartig et al. 2003, J Environ Psychol, 23; 109-123). I understand and accept that the author belong to the green exercise tradition and physical activity is the starting point for their study. However, there are no measures on how the participants perceive the environment’s impact on their physical activity performance during the trip, only on their perceived well-being.

In light of this, I recommend that the author reconsider the title of the manuscript to better reflect the main message of the results. “Add-on-effect” is only used in the title and not in the text itself. I do not understand what is adding on to what since the study design does not allow for differentiation between psychological benefits of the physical activity itself and the impact of the visual environment. I my view, the study is all about the impact of anthropogenic elements in the environments on affective states and stress during mountain hiking, or rather, does it matter if human made elements is added to nature environments when it comes to the potential for such benefits.

The value of the data testing the second aim is questionable since the participants have a high average score on NRS. This is only mentioned as a limitation but should be pointed out in the premises for the choices described earlier in the manuscript or as a part of the results section. The high NRS score reported by the participants limits the relevance of the results to this particular group. NRS is often used as a covariate in analysis. Table 1 also indicate that there is variation in level of physical activity and weekly outdoor trips in the material.

The study applies a cross over design, which is a strength of the study. However, as the author discuss there are many challenges with such field studies, which they also acknowledge in the discussion. The authors have also carefully selected tracks and exposures to make the conditions comparable in light of physical demands and type of landscape. Even then it would be of interest to know what type of landscape the study is located in. Anthropogenic elements in nature can be many different and they can give different associations, and thereby possibly different impacts. The study must have been taken through during the summer time, and some of the anthropogenic element mentioned can give negative impressions and some may have given positive indications of joyful activities offered during other times of the year.

I am curious about the social environment during the study and how the group dynamics may have influenced the study. Obviously, the trip has been taken through in groups of more than 25 people. And we do not know how they have interacted and discussed their experiences between the trips. It also seems that some of the participants were related?

Conclusion; more commenting on the benefits of hiking itself than to relate to the aims of the study. Are the data supporting any impact of mountain hiking on cortisol levels – the design does not open for such claims in the conclusion?

Minor points:

Line 136: Is Physical activity readiness the same as “self reported physical activity” (line 197) measured with the International Physical Activity Quest.?

Line 241: Where is the outcome of the walking intensity measures reported? If it is only to describe the intensity, it could be reported under 2.4.1?

Line 205; “Four different sets of self-reported questionnaires based on the circumplex model…” The two first questionnaires is easily to relate to the circumplex model. Please explain how the next two “sets” are related to the circumplex model.

In general, there is not always that the order of different components in the text is the same. Examples; The abstract mention acute stress-related physiological responses before the affective states while the order is the opposite later in the text. The variables in Table 1 is not the same as the order mentioned in 2.5.1. The variables in Table 2 are not in the same order as in the methods section.

Author Response

At first, we want to thank the Reviewers for their time spent on the evaluation of the manuscript. We believe the comments greatly helped to improve the manuscript.

Please find the comments of the Reviewers in black and the author’s responses in blue.

Reviewer #2

Comments and Suggestions for Authors

Review of the manuscript «Is there an environment-based add-on-effect of mountain hiking on affective states and cortisol concentration? A cross over trail.

Thank you for the opportunity to read the manuscript. The manuscript has a good structure and provides a lot of useful information about the field study. I have a few comments that may help discussing their approach and improve the text.

We want to thank the Reviewer for that comment.

The manuscript contributes to the discussion on how environmental qualities may promote or enhance psychological benefits of mountain hiking as a form of physical activity. The design and exposure conditions selected may also be viewed as a possibility to uncover to what extent anthropogenic components in the environment play a role for the psychological benefits of being active in nature. It is worth mentioning that mountain hiking was an exposure of interest in the early classic studies on nature exposure in environmental psychology (see Hartig et al. 2003, J Environ Psychol, 23; 109-123).

Thank you for the citation. Although not explicitly stated in Hartig et al. (2003) that mountain hiking (i.e. walking up and down with altitude differences) was applied, we agree that the paper fits very well to the topic and included the citation in the introduction as following:

“Walking in mountainous areas was used earlier to study restorative effects of the environment on attention, emotion, and blood pressure [17]. Thereby, mountain hiking can be considered an appropriate form of physical activity to study possible influences of anthropogenic elements on affective states.”

I understand and accept that the author belong to the green exercise tradition and physical activity is the starting point for their study. However, there are no measures on how the participants perceive the environment’s impact on their physical activity performance during the trip, only on their perceived well-being.

In light of this, I recommend that the author reconsider the title of the manuscript to better reflect the main message of the results. “Add-on-effect” is only used in the title and not in the text itself. I do not understand what is adding on to what since the study design does not allow for differentiation between psychological benefits of the physical activity itself and the impact of the visual environment. I my view, the study is all about the impact of anthropogenic elements in the environments on affective states and stress during mountain hiking, or rather, does it matter if human made elements is added to nature environments when it comes to the potential for such benefits.

Thank you. We agree with the considerations and changed the title into:

“The role of anthropogenic elements in the environment for affective states and cortisol concentration in mountain hiking - A crossover trial”

With the previous title, we wanted to state that - according to the green exercise tradition - environmental effects possibly add on to exercise effects. However, we agree that the title in its present form is reflecting the content of the manuscript more adequately.

The value of the data testing the second aim is questionable since the participants have a high average score on NRS. This is only mentioned as a limitation but should be pointed out in the premises for the choices described earlier in the manuscript or as a part of the results section. The high NRS score reported by the participants limits the relevance of the results to this particular group. NRS is often used as a covariate in analysis. Table 1 also indicate that there is variation in level of physical activity and weekly outdoor trips in the material.

Thank you for the possibility to defend on our approach. We agree that the “value of the data testing the second aim is questionable” and that a series of analyses of covariance (ANCOVA) is an analysis option, when the effect of nature relatedness should be controlled for in the analysis. However, we did not change the statistical analysis in the present version of the manuscript for the following reasons:

1) Our analysis strategy with the median NRS value was determined a priori to the study.

2) The main goal of an ANCOVA is to “partial out the influence of the covariate” on the dependent variable (Field, 2009, page 396). Our initial hypothesis was “that persons with higher nature relatedness might benefit to a greater extent from green exercise in an environment with less anthropogenic elements compared to persons with lower nature relatedness” (opposed to partial out the effect of the covariate). The 2 × 2 × 2 mixed-model ANOVA might be considered an appropriate way to test this hypothesis, since we did not want to partial out the effect of NRS but test the mentioned hypothesis.  

3) We that do not agree with the Reviewers comment that the “high average score on NRS … is only mentioned as a limitation” since we discussed the high NRS score in the discussion section where we considered as adequately (i.e. in the section on 4.3. Nature Relatedness opposed to the 4.4 Limitations section). We believe that the limited value of data is discussed appropriately (“Thereby, the present sample is considered a homogenous group with high nature relatedness, what might have concealed possible influences of nature relatedness.” And “Thereby, the possible influence of nature relatedness requires further research.”).

4) A series of ANCOVAs does not eliminate the high nature relatedness of the sample.

To mention the limitation of the high nature relatedness earlier in the manuscript and following your suggestion we included the following sentence to the statistical analysis section, where the analysis based on the median value was specified:

 “It should be noted that median nature relatedness was high in the present sample (4.1) compared to previous studies on nature relatedness (mean value of environmentalists 3.7 [17]). Since our statistical analysis approach was determined a priori we continued with the analysis strategy.”

We conducted a series of ANCOVAs with the following results for the time*condition interaction.

p-value   time*condition

Affective   valence

0.153

Arousal

0.546

Activation

0.042

Elation

0.277

Calmness

0.005

Fatigue

0.625

Depression

0.011

Contemplation

0.061

Anger

0.042

Excitement

0.728

Anxiety

0.190

Cortisol

0.092

Accordingly, the time*condition interaction gets significant for the subscale calmness indicating increased calmness after N compared to decreased calmness after H when the effect of nature relatedness is controlled for. However, this analysis also does not eliminate the high nature relatedness of the sample. Consequently, our conclusion (“the possible influence of nature relatedness requires further research”) may similarly remain.

However, we are happy to present the results of the ANCOVA instead, if the Reviewer is not satisfied with our argumentation.

The study applies a cross over design, which is a strength of the study. However, as the author discuss there are many challenges with such field studies, which they also acknowledge in the discussion. The authors have also carefully selected tracks and exposures to make the conditions comparable in light of physical demands and type of landscape. Even then it would be of interest to know what type of landscape the study is located in. Anthropogenic elements in nature can be many different and they can give different associations, and thereby possibly different impacts. The study must have been taken through during the summer time, and some of the anthropogenic element mentioned can give negative impressions and some may have given positive indications of joyful activities offered during other times of the year.

If we understand this comment of the Reviewer correctly, a more detailed description of the landscape where the study was conducted in is suggested. Following also the comments of Reviewer #1, we included a figure showing the typical view in the environment with less anthropogenic elements (N) and the environment with more anthropogenic elements (H).

Figure 3. Typical view while hiking in the environment with less anthropogenic elements (N, two upper pictures) and the environment with more anthropogenic elements (H, three lower pictures).

The Reviewer assumed that the study was conducted in summer time. As it was stated in the previous version of the manuscript, the study was conducted in October, what is referring to mid-autumn in Austria. We added a more detailed description to the design section of the manuscript:

“The experimental phase was conducted on three consecutive days in October 2017 (mid-autumn) and was located in a renowned summer and winter sports area in Austria.”

Thank you for mentioning that “some of the anthropogenic element mentioned can give negative impressions and some may have given positive indications of joyful activities offered during other times of the year”. We agree on that point and included it in the discussion section:

“Another possible explanation for the lack favorable effects in the environment with less anthropogenic elements might refer to the anthropogenic elements ski lifts and snow cannons. In winter time, potentially joyful activities like skiing and snowboarding are realized by ski lifts and snow cannons. Consistent with mood congruency [37], recall on previous joyful activities might have influenced affective responses in the present study.”

I am curious about the social environment during the study and how the group dynamics may have influenced the study. Obviously, the trip has been taken through in groups of more than 25 people. And we do not know how they have interacted and discussed their experiences between the trips. It also seems that some of the participants were related?

We agree that social influences are important in the assessment of the outcomes of the present study. The Reviewer correctly assumed that some of the participants were related and that the hiking tours were conducted in the total subgroup of 25 and 27 participants, what can be considered as similar to guided mountain hiking tours in the Austrian Alpine Association. The participants were allowed to discuss their experiences between the two hikes since we considered it as impossible to control for or ban such discussions.

The following steps regarding the psychological measures were conducted to minimize social influences during the study:

1) Measurements pre and post intervention were conducted and changes were analyzed to control for differences pre mountain hiking (e.g. triggered by social interaction between the hikes).

2) The subgroups stayed identical on tour 1 and 2.

3) Regarding the social interactions between research team and participants, the team of researchers per group did not change between the two days. We did not consider hiking alone for approximately three hours as a realistic and practical option.

The following analysis suggests that social interactions (or other sources of influence) between the first and the second hiking day played a minor role in the main outcome change in affective responses in the present study design. If we assume that social interactions might affect both subgroups similarly, we would expect differences in pre-post changes in affective responses between the first and the second hiking day, i.e. “period effects” following the nomenclature of Senn 2002. We analyzed period effects using a 2x2 mixed-model ANOVA with group (1: first N, then H; 2: first H, then N) and period (1: first hiking day, 2: second hiking day) and did not find significant period effects, when Bonferroni correction was applied.

period

Δ   Affective valence

0.192

Δ   Arousal

0.038

Δ   Activation

0.490

Δ   Elation

0.025

Δ   Calmness

0.557

Δ   Fatigue

0.201

Δ   Depression

0.063

Δ   Contemplation

0.388

Δ   Anger

0.031

Δ   Excitement

0.854

Δ   Anxiety

0.159

However, since we cannot fully rule out the influence of social interaction in the study, we provided another limitation in the limitations section, which now reads as:

“Fourthly, we did not control for social interaction between the participants what might have affected the results.”

Conclusion; more commenting on the benefits of hiking itself than to relate to the aims of the study. Are the data supporting any impact of mountain hiking on cortisol levels – the design does not open for such claims in the conclusion?

This is a valid point. We admit that we focused extensively on the integration of our previous finding on mountain hiking in the conclusion section. We adapted the conclusion section as following and focused more strongly on the findings of the present study.

“Conclusions

The present field study focused on a possible impact of anthropogenic elements in the environment on affective states and salivary cortisol level in a single three-hour bout of green exercise (mountain hiking). According to the present results, we conclude that anthropogenic elements in the natural environment seem to be a minor influencing factor on the change of the outcomes observed. The role of nature relatedness remains unclear and future research is warranted.”

Minor points:

Line 136: Is Physical activity readiness the same as “self reported physical activity” (line 197) measured with the International Physical Activity Quest.?

We are sorry for the misunderstanding, which probably resulted on the fact, that we did not display the Physical Activity Readiness Questionnaire in the methods section (including Figure 1).

The Physical Activity Readiness Questionnaire is not the same as the International Physical Activity Questionnaire. The Physical Activity Readiness Questionnaire was used for assessing the eligibility of the participants; while the International Physical Activity Questionnaire IPAQ was used for the quantification of physical activity level. We adapted Figure 1 and included the following sentences in the initial screening section:

“Information on the ability to be physically active, sociodemographic data, physical activity, nature relatedness and perceived stress was collected in a web-based questionnaire with 74 questions (total time to complete: approximately 30 minutes). The six-item Physical Activity Readiness Questionnaire [20] was used for assessing the ability to be physically active.”

Line 241: Where is the outcome of the walking intensity measures reported? If it is only to describe the intensity, it could be reported under 2.4.1?

We apologize for not stating walking intensity explicitly in the results section, where we only referred to “average heart rate”. We adapted the results section as follows:

“Walking intensity according to the average heart rate (n = 10) was comparable in N (63 (10) % of age-predicted maximal heart rate) and H (63 (7) % of age-predicted maximal heart rate), respectively.”

Line 205; “Four different sets of self-reported questionnaires based on the circumplex model…” The two first questionnaires is easily to relate to the circumplex model. Please explain how the next two “sets” are related to the circumplex model.

Abele-Brehm et al. (1986), the authors of the first categorical questionnaire (Mood Survey Scale) provide the following figure of the subscales located in the Circumplex model (the authors provided the figure in German; therefore we adapted the figure using the English terms)

According to Russel (1980), the term “afraid” is located in the upper left quadrant. The subscale state anxiety is considered more negatively and with lower arousal colored compared to “excitement”, but more positively and with higher arousal compared to “anger”.

We included the following clarifications to the manuscript:

“The subscales can be located in the Circumplex model: activation and excitement mark affective states with high arousal. Activation is characterized with more positive valence and excitement with more negative valence. The subscales fatigue (negative valence) and calmness (positive valence) mark low arousal. Anger, depression (both negative valence), contemplation (neutral valence) and elation (positive valence) all mark medium aroused states [33].”

“In the Circumplex model, anxiety is described with higher arousal and more negative valence [6].”

In general, there is not always that the order of different components in the text is the same. Examples; The abstract mention acute stress-related physiological responses before the affective states while the order is the opposite later in the text. The variables in Table 1 is not the same as the order mentioned in 2.5.1. The variables in Table 2 are not in the same order as in the methods section.

Thank you; this is a valid point. We adapted the manuscript that the order is identical in the Material and Methods and Results section. For the outcomes, the following order was used consistently: dimensional affective responses, categorical affective responses, salivary cortisol concentration. For the initial screening variables, the following order was used: ability to be physically active, sociodemographic data, physical activity, nature relatedness and perceived stress.

However, in Table 2, categorical sociodemographic variables were displayed separately after all continuous variables to avoid switching between mean (standard deviation) and % (n) in the table.

Again, we want to thank the Reviewers for their time and effort.

Round 2

Reviewer 1 Report

Physiological indicators such as cortisol are affected by the environment such as climate rather than the landscape.

People relax in the natural environment even if there are anthropogenic elements.

So, I think to be better to experiment with constant conditions in the laboratory .